# XAttention: Block Sparse Attention with Antidiagonal Scoring

**Ruyi Xu** [* 1]  **Guangxuan Xiao** [* 2]  **Haofeng Huang** [1]  **Junxian Guo** [3]  **Song Han** [2 4]

https://github.com/mit-han-lab/x-attention

## Abstract

Long-Context Transformer Models (LCTMs) are vital for real-world applications but suffer high computational costs due to attention's quadratic complexity. Block-sparse attention mitigates this by focusing computation on critical regions, yet existing methods struggle with balancing accuracy and efficiency due to costly block importance measurements. In this paper, we introduce XAttention, a plug-and-play framework that dramatically accelerates long-context inference in Transformers models using sparse attention. XAttention's key innovation is the insight that the sum of antidiagonal values (i.e., from the lower-left to upper-right) in the attention matrix provides a powerful proxy for block importance. This allows for precise identification and pruning of non-essential blocks, resulting in high sparsity and dramatically accelerated inference. Across comprehensive evaluations on demanding long-context benchmarks—including RULER and LongBench for language, VideoMME for video understanding, and VBench for video generation—XAttention achieves **accuracy comparable to full attention** while delivering substantial computational gains. We demonstrate up to **13.5×** **acceleration** in attention computation. These results underscore XAttention's ability to unlock the practical potential of block sparse attention, paving the way for scalable and efficient deployment of LCTMs in real-world applications.

## 1. Introduction

The transformative impact of Large Language Models (LLMs) (Dubey et al., 2024; OpenAI, 2023) is expanding beyond natural language processing, steering in a new era of multimodal capabilities. Long-Context Transformer Models (LCTMs) are emerging as essential tools in this evolution, particularly for tasks like video understanding (Lin et al., 2023; Wang et al., 2024) and video generation (Kong et al., 2025) that demand processing and generating exceptionally long sequences of information. These models hold the key to unlocking brilliant systems capable of interacting with the world in a human-like way, understanding and generating not just text, but also visual information over extended periods. Imagine AI agents engaging in seamless, multimodal, day-long interactions, or powerful world simulators generating hours of coherent video—tasks that hinge on processing a tremendous number of tokens.

However, realizing this vision requires overcoming a significant challenge: the computational burden of the attention mechanism (Vaswani et al., 2017). While crucial for capturing relationships within sequences, attention's cost scales quadratically with sequence length. This quadratic scaling creates a substantial bottleneck during the pre-filling stage, hindering the practical deployment of LCTMs for complex, real-world applications.

In the pursuit of more efficient Transformers, block-sparse attention (Zaheer et al., 2020; Guo et al., 2024) has emerged as a promising avenue. The core idea is appealing: instead of computing attention between all token pairs, focus resources on the most crucial regions of the attention map, creating "blocks" of relevant information. This selective computation promises to drastically reduce computational burden while preserving the model's ability to capture essential long-range dependencies.

Yet, existing block-sparse methods have struggled to deliver on their full potential, often grappling with a trade-off between accuracy and efficiency. This stems from the lack of lightweight yet effective mechanisms for identifying and prioritizing truly important attention blocks. The overhead involved in determining block importance can negate the gains achieved through sparsity, rendering these methods impractical for real-world deployment.

This leads us to a question: *Can we design a block-sparse attention mechanism that dramatically accelerates long-context Transformers without compromising accuracy, truly unlocking their potential for real-world applications?*

---

[*]Equal contribution  [1]Tsinghua University [2]Massachusetts Institute of Technology [3]SJTU [4]NVIDIA. Correspondence to: Guangxuan Xiao <xgx@mit.edu>, Song Han <songhan@mit.edu>.

*Proceedings of the 42nd International Conference on Machine Learning*, Vancouver, Canada. PMLR 267, 2025. Copyright 2025 by the author(s).

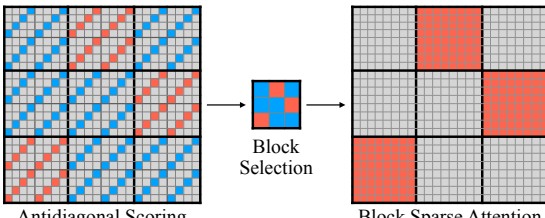
Antidiagonal Scoring     Block Selection     Block Sparse Attention

*Figure 1.* **Illustration of XAttention:** XAttention optimizes attention through a three-step process: (Left) Strided Antidiagonal Scoring: Each block (8×8 in this example) is scored by summing values along its strided antidiagonals (stride = 4), with red lines generally indicating higher summed values and blue lines lower values. (Middle) Block Selection: High-scoring blocks are selected based on these evaluations. (Right) Block Sparse Attention: Attention is computed only on the selected blocks (red blocks on the right), achieving substantial computational savings. This example uses a sequence length of 24.

We answer this question by introducing XAttention, a novel plug-and-play framework designed to significantly improve the efficiency of block-sparse attention in long-context Transformers. XAttention is based on the key insight that the sum of antidiagonal values within the attention matrix can serve as a powerful, yet computationally efficient, indicator of block importance. Unlike existing methods that primarily rely on computationally intensive and lossy solutions like token pooling to identify important blocks, XAttention leverages this simple score to offer a potentially more streamlined and direct approach for rapidly and accurately identifying critical attention blocks.

This antidiagonal scoring algorithm allows XAttention to aggressively find and prune non-essential computations, achieving substantial sparsity without sacrificing accuracy. We extensively evaluate XAttention on challenging long-context benchmarks, including RULER and LongBench for natural language processing, VideoMME for video understanding, and VBench for video generation. Across these benchmarks, XAttention achieves accuracy comparable to full attention while delivering substantial computational gains, demonstrating up to 13.5× acceleration in attention computation during pre-filling. These results underscore XAttention's ability to unlock the practical potential of block-sparse attention, paving the way for scalable and efficient deployment of long-context Transformers in demanding applications, especially in the expanding field of multimodal AI.

## 2. Method

In this section, we introduce our method, **XAttention**. The XAttention algorithm comprises three primary components: (1) importance prediction of attention map blocks, (2) selection of important attention blocks, and (3) prediction of the minimum threshold for attention heads.

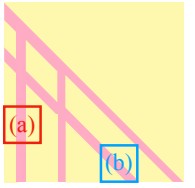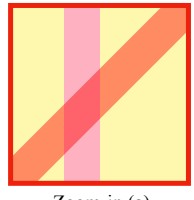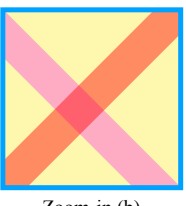
Vertical-Slash Pattern     Zoom-in (a)     Zoom-in (b)

*Figure 2.* XAttention's antidiagonal pattern intersects both vertical and slash patterns within a block, enabling efficient detection of these patterns and guiding effective sparse attention computation.

### 2.1. Importance Prediction

The inherent sparsity of attention maps necessitates a robust strategy for predicting the importance of attention blocks. While methods like MInference (Jiang et al., 2024) and FlexPrefill (Lai et al., 2025) utilize a combination of pooling and "vertical slash detection," our ablation study reveals that relying solely on average or sum pooling yields inaccurate predictions. Pooling methods are particularly ineffective when only a few significant vertical or slash patterns exist within a block, failing to capture these crucial indicators of importance.

MInference and FlexPrefill attempt to overcome this limitation by analyzing the last segment of the input query to identify important "vertical and slash indices." However, this approach faces two key challenges: firstly, important attention patterns may not persist in the final query segment; secondly, the search algorithm itself introduces substantial computational overhead (demonstrated in Figure 6).

Fundamentally, an effective block importance prediction method should automatically and robustly identify significant patterns, including crucial vertical and slash patterns. To achieve this, we propose the **antidiagonal selection method**. Within each block of size $B$, we select elements along the antidiagonal using a stride $S$ (visualized in Figure 1). The sum of these selected elements serves as a proxy for the overall importance of the corresponding attention block.

The effectiveness of this method can be understood from two perspectives: (1) *Information Preservation*: This selection strategy ensures that information from all tokens is considered, as each token contributes to at least one antidiagonal sum. (2) *Pattern Detection:* As illustrated in Figure 2, the antidiagonal intersects every possible vertical and slash pattern within a block. XAttention's antidiagonal pattern intersects both vertical and slash patterns within a block, enabling efficient detection of these patterns and guiding effective sparse attention computation. This ensures that no crucial patterns are missed during the importance estimation process.

## 2.2. Threshold Block selection

Based on the antidiagonal scoring pattern, we propose the following sparse attention block selection algorithm. Let $S$ denote the stride, and let $B$ be the size of the sparse attention blocks. The process begins with *antidiagonal summation*, where we select elements along the antidiagonal within each $S \times S$ block of the attention map and compute the sum of these elements for each antidiagonal. Subsequently, we perform *softmax normalization* by applying the softmax function to these antidiagonal sums, yielding a probability distribution over the antidiagonals. Finally, for *block selection*, the `find_blocks` function is employed to identify the minimal set of blocks whose cumulative sum of antidiagonal probabilities exceeds a predefined threshold $\tau$. Formally, this can be expressed as:

$$\mathtt{find\_blocks}(A, \tau) = \arg\min_{\mathcal{B}} \left\{ |\mathcal{B}| \ \Big| \ \sum_{b \in \mathcal{B}} \sum_{(i,j) \in b} A_{i,j} \geq \tau \right\}$$

where $A$ is the attention map, $\mathcal{B}$ is a set of blocks, and $|\mathcal{B}|$ represents the number of blocks in the set. This process effectively determines the most important blocks in the attention map based on the antidiagonal scoring pattern and the specified threshold.

---

**Algorithm 1** Block Selection

---

**Require:** Query matrix $Q \in \mathbb{R}^{L \times d}$, Key matrix $K \in \mathbb{R}^{L \times d}$, block size $B$, stride $S$, head dimension $d_h$, threshold $\tau$
**Ensure:** Sparse mask $M$
1: $N_B \leftarrow \lfloor L/B \rfloor$ {Number of blocks}
2: **for** $b = 0$ to $N_B - 1$ **do**
3:    $Q_{\text{slice}} \leftarrow Q[bB : (b+1)B, :]$ {Extract $Q$ block}
4:    $Q_{\text{reshaped}} \leftarrow []$
5:    **for** $i = S - 1$ down to $0$ **do**
6:       $Q_{\text{reshaped}}.\text{append}(Q_{\text{slice}}[i :: S, :])$ {Reshape along antidiagonals with stride $S$}
7:    **end for**
8:    $K_{\text{reshaped}} \leftarrow []$
9:    **for** $i = 0$ to $S - 1$ **do**
10:      $K_{\text{reshaped}}.\text{append}(K[i :: S, :])$ {Reshape along antidiagonals with stride $S$}
11:    **end for**
12:    $A_{\text{approx}} \leftarrow \text{Softmax}\left(\frac{Q_{\text{reshaped}} K_{\text{reshaped}}^T}{\sqrt{d_h} \cdot S}\right)$ {Approximate attention scores}
13:    $M_b \leftarrow \text{find\_blocks}(A_{\text{approx}}, \tau)$ {Find blocks based on threshold}
14: **end for**
15: $M \leftarrow \text{concatenate}(M_0, M_1, \ldots, M_{N_B-1})$ {Concatenate block masks}

---

## 2.3. Minimum Threshold Prediction

We propose a dynamic programming approach to determine the optimal threshold for each attention head. Previous research indicates that different attention heads exhibit varying sparsity levels and importance. Thus, it is beneficial to dynamically adjust thresholds for individual heads to optimize the balance between accuracy and computational efficiency.

**Problem Formulation:** Consider a model with $H$ attention heads. We define a dynamic programming table $D[h][m]$, where $h \in \{1, 2, \ldots, H\}$ represents the $h$-th head, and $m \in \{1, 2, \ldots, M\}$ denotes the number of threshold adjustments made. $D[h][m]$ stores the best performance achievable when exactly $m$ threshold adjustments have been made across the first $h$ heads.

**Dynamic Programming:** Our objective is to find the optimal threshold for each head such that their joint contribution maximizes accuracy while minimizing computation. The recurrence relation for the DP table is:

$$D[h][m] = \max(D[h-1][m], P(h, m))$$

where $P(h, m)$ represents the performance of the model when the $h$-th head's threshold is adjusted for the $m$-th time. This corresponds to the model's performance after reducing the threshold of the $h$-th head by one step relative to the state $D[h-1][m-1]$ in the optimization process.

We adjust the threshold for each head by reducing it by 10% at each step:

$$t_h(m) = t_h(m-1) \times 0.9$$

This ensures a gradual reduction in computation while preserving each head's contribution to accuracy.

Note that this dynamic threshold prediction method can further optimize XAttention's sparsity but is not a mandatory component. We present detailed results in the ablation study.

# 3. Experiments

This section presents our empirical investigation into the effectiveness of XAttention. We first detail the implementation specifics, followed by evaluation results on text and video understanding, as well as video generation benchmarks, against strong baselines. We then test the acceleration performance of XAttention. Finally, we provide analytical ablation studies to further understand the behavior of XAttention.

## 3.1. Experimental Setup

**Models** We evaluate XAttention across three distinct domains. For natural language tasks, we employ Llama-3.1-8B-Instruct (Dubey et al., 2024). In the video understanding

*Table 1.* Accuracy comparison of different methods and sequence lengths on RULER with Llama-3.1-8B-Instruct . XAttention is configured with Stride $S = 8$ and $S = 16$ with Precisely Predicted Minimum Threshold.

| Input Len | 4k | 8k | 16k | 32k | 64k | 128k | Avg. |
|---|---|---|---|---|---|---|---|
| Full | 96.74 | 94.03 | 92.02 | 84.17 | 81.32 | 76.89 | 87.52 |
| FlexPrefill | 95.99 | 93.67 | 92.73 | 88.14 | 81.14 | **74.67** | 87.72 |
| MInference | 96.54 | 94.06 | 91.37 | 85.79 | 83.03 | 54.12 | 84.15 |
| SeerAttn | 95.32 | 92.14 | 92.20 | 88.05 | 83.30 | 72.37 | 87.23 |
| Xattn S=8 | **96.83** | **94.07** | 93.17 | **90.75** | **84.08** | 72.31 | **88.47** |
| Xattn S=16 | 96.11 | 93.95 | **93.56** | 90.64 | 83.12 | 71.11 | 88.08 |

domain, we utilize Qwen2-VL-7B-Instruct (Wang et al., 2024). Finally, for video generation, we use the Hunyuan-Video model (Kong et al., 2025). To optimize the trade-off between computational efficiency and accuracy on natural language tasks, we apply our precise threshold prediction method to the Llama-3.1-8B-Instruct model.

**Baselines** We compare XAttention against several strong baselines. Our primary baseline for dense attention is FlashAttention (Dao, 2023), implemented within the Flash-Infer (Ye et al., 2024) framework. We also compare against MInference (Jiang et al., 2024), FlexPrefill (Lai et al., 2025), and SeerAttention (Gao et al., 2024), strictly adhering to their public implementations. For SeerAttention, we incorporate pretraining on the Gare weights. For MInference, we utilize their official configuration, where all attention heads adopt the "Vertical-Slash" sparsity pattern. For FlexPrefill, we set the hyperparameters to $\gamma = 0.95$ and $\tau = 0.1$, which, according to the original paper, resulted in the highest accuracy among the provided parameter sets.

**Datasets** We evaluate our model on a diverse set of tasks spanning natural language understanding, video understanding, and video generation. For natural language tasks, we employ the RULER (Hsieh et al., 2024) dataset, a synthetic benchmark specifically designed to assess long-context abilities in LLMs. RULER allows for customizable sequence lengths and task complexities, extending the traditional needle-in-a-haystack test while introducing novel task categories like multi-hop tracing and aggregation. We also evaluate on real-world long-context tasks from LongBench (Bai et al., 2023) to test performance in practical scenarios.

For video understanding, we utilize the Video-MME (Fu et al., 2024) dataset, the first comprehensive benchmark for evaluating multimodal large language models (MLLMs) on video analysis. Video-MME comprises 900 videos totaling 254 hours, with durations ranging from 11 seconds to 1 hour, providing a robust testbed for assessing long video comprehension.

In the video generation domain, we leverage 946 GPT-

augmented text prompts from VBench (Huang et al., 2024) to generate videos. We then compare the videos generated by our proposed method, XAttention, against those produced by a full attention baseline, evaluating the effectiveness of our approach in generating high-quality video content.

### 3.2. Accuracy Results

**RULER** On the RULER benchmark (Hsieh et al., 2024), we apply the dynamic programming method described in Section 3.3 for Minimum Threshold Prediction, utilizing strides of $S = 8$ and $S = 16$ with a maximum adjustment number of $M = 1000$. This yielded a set of minimum thresholds with an average of 0.8, further enhancing the computational efficiency of our sparse attention mechanism.

Table 1 compares the accuracy of XAttention against strong baselines on the Llama-3.1-8B-Instruct model across various sequence lengths on RULER. Notably, both MInference and SeerAttention experience significant performance degradation as context length increases. In contrast, XAttention, configured with $S = 8$ and $S = 16$ and employing our precisely predicted minimum thresholds, not only surpasses the optimal sparse attention baseline, FlexPrefill, but also outperforms full attention at several sequence lengths. Additionally, we evaluate the same tasks on the above three models. The results shown in appendix are consistent across models.This demonstrates the robustness of XAttention in handling very long contexts.

**LongBench** Table 2 presents the performance of XAttention compared to strong baselines on the real-world tasks within the LongBench benchmark, using the Llama-3.1-8B-Instruct model. Maintaining the same configuration used for the RULER evaluation, we evaluate XAttention alongside MInference and FlexPrefill. XAttention achieves the highest average score across all tasks, demonstrating its effectiveness in practical scenarios. Notably, the performance of XAttention on individual tasks remains close to that of full attention, indicating that our method preserves accuracy while improving efficiency.

**Video Understanding** We apply Stride $S = 16$ and threshold $\tau = 0.9$ parameters on the QwenVL-2-7B model. As shown in Table 3, among the three sparse attention methods, MInference and FlexPrefill fail to achieve optimal performance on Long video tasks. XAttention achieves the best average score among all sparse attention methods and even outperforms FlashAttention on long videos, with a frame rate of 1 frame per second for up to 1 hour.

**Video Generation** We evaluate XAttention's performance in the video generation domain using the HunyuanVideo model on prompts from VBench (Huang et al., 2024). The

*Table 2.* Comparison of different attention methods on real-world LongBench tasks using the Llama-3.1-8B-Instruct model. XAttention, configured with stride 8 and Precisely Predicted Minimum Threshold, achieves the best average scores against all baselines.

| Method | Single-Doc QA | | | Multi-Doc QA | | | Summarization | | | | Few-shot Learning | | | Code | | | Avg. |
| | NrtvQA | Qasper | MF-en | HPQA | 2WikiMQA | MuSiQue | GovReport | QMSum | VCSum | MultiNews | TREC | TriviaQA | SAMSum | LSHT | LCC | RB-P | |
|---|---|---|---|---|---|---|---|---|---|---|---|---|---|---|---|---|---|
| Full | 31.44 | 25.07 | 29.40 | 16.89 | 17.00 | 11.79 | 34.22 | 23.25 | 15.91 | 26.69 | 72.50 | 91.65 | 43.74 | 46.00 | 52.19 | 49.14 | 40.34 |
| MInference | **31.59** | 24.82 | 29.53 | 17.03 | 16.46 | 11.58 | 34.19 | 23.06 | **16.08** | 26.71 | **72.50** | 91.18 | 43.55 | 46.00 | 52.33 | 49.93 | 40.30 |
| FlexPrefill | 27.30 | **28.56** | 27.66 | 17.20 | 15.14 | 9.46 | 32.76 | **23.66** | 16.05 | **27.25** | 64.00 | 88.18 | 41.28 | 31.00 | 45.69 | 47.54 | 36.83 |
| XAttention | 28.99 | 26.14 | **29.92** | **17.40** | **16.70** | **11.80** | **34.41** | 23.26 | 16.00 | 27.04 | 72.00 | **91.65** | **43.86** | **47.00** | **52.67** | **50.84** | **40.60** |

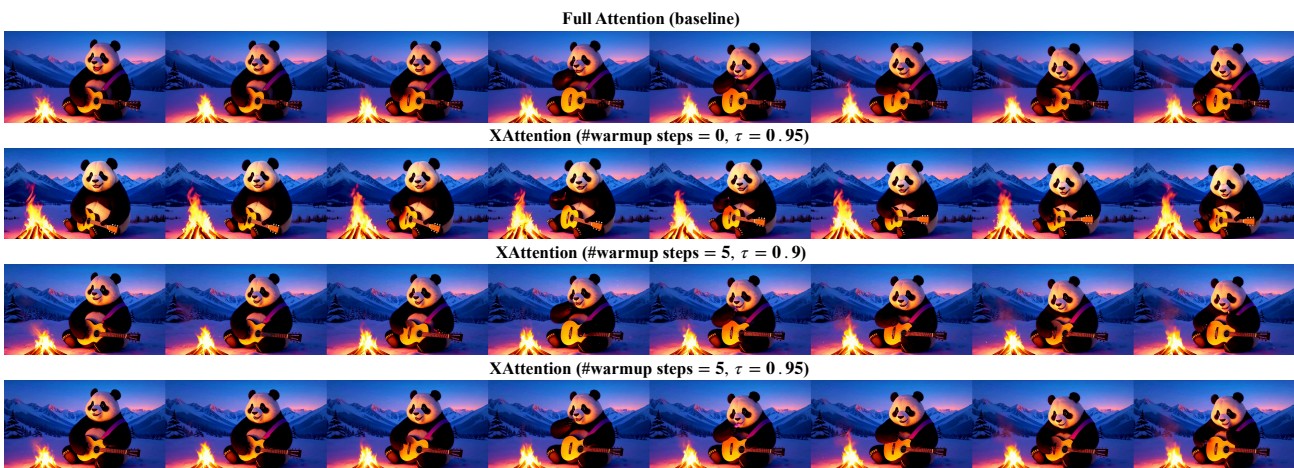

**Full Attention (baseline)**

**XAttention (#warmup steps = 0, $\tau = 0.95$)**

**XAttention (#warmup steps = 5, $\tau = 0.9$)**

**XAttention (#warmup steps = 5, $\tau = 0.95$)**

**Prompt**: *"A joyful, fuzzy panda sits cross-legged by a crackling campfire, strumming a small acoustic guitar with enthusiasm. The panda's black and white fur contrasts beautifully with the warm glow of the fire, casting flickering shadows on the surrounding snow-covered ground. Behind the panda, majestic snow-capped mountains rise against a twilight sky, their peaks tinged with the last light of the setting sun. The panda's eyes sparkle with delight as it plays a cheerful tune, the serene mountain landscape and the cozy campfire creating a magical, heartwarming scene."*

*Figure 3.* Qualitative comparison of video generation results on the VBench benchmark using the first prompt in the VBench dataset. Rows show frames from videos generated using: (1) Full Attention (baseline), (2) XAttention with no warmup and ($\tau = 0.95$), (3) XAttention with 5 warmup steps and ($\tau = 0.9$), and (4) XAttention with 5 warmup steps and ($\tau = 0.95$). XAttention with warmup achieves high visual fidelity to the full attention baseline.

HunyuanVideo model utilizes the Diffusion Transformer (DiT) architecture (Peebles & Xie, 2023), which employs non-causal attention. As existing baselines are not implemented for non-causal attention, we compare XAttention solely against the full attention baseline. Our evaluation considers both quantitative metrics (PSNR, SSIM, LPIPS) and qualitative visual comparisons. We replace all attention computations in the DiT backbone with XAttention, and measure performance against the full attention output using the same random seed and prompt, averaging the results across all 946 VBench prompts. The generated videos have a resolution of 720×1280 pixels and 129 frames, with 50 denoising steps. We configure XAttention with a stride of $S = 8$ and thresholds of $\tau = 0.9$ and $\tau = 0.95$.

Initially, applying XAttention from the very beginning of the denoising process in the HunyuanVideo model led to slight layout shifts in the output video compared to the full attention baseline, resulting in lower quantitative scores. Inspired by research on diffusion models (Xiao et al., 2023c; Li et al., 2024) demonstrating that early denoising steps are critical for determining content layout, we introduce a "warmup" phase. During this phase, we utilize full attention for the first 5 denoising steps, before switching to XAttention. Figure 3 illustrates the qualitative impact of this warmup strategy.

Table 4 presents the quantitative results of applying XAttention to the HunyuanVideo model. Both configurations, with thresholds of $\tau = 0.90$ and $\tau = 0.95$, achieve high fidelity compared to videos generated with full attention. Specifically, we observe a PSNR up to 23.5, SSIM up to 0.822, and LPIPS down to 0.155, indicating a level of similarity that is difficult for the human eye to discern. As expected, a

*Table 3.* Comparison of different methods on QwenVL-2-7B in the Video-MME video understanding task. XAttention is configured with Stride $S = 16$ and Threshold $\tau = 0.9$. XAttention outperforms Full Attention on long video tasks and achieves the best average performance among all sparse attention methods.

| | Short (%) | | Medium (%) | | Long (%) | | Overall (%) | |
|---|---|---|---|---|---|---|---|---|
| subs | w/o | w/ | w/o | w/ | w/o | w/ | w/o | w/ |
| Full | 72.1 | 78.1 | 63.9 | 69.4 | 55.1 | 60.2 | 63.7 | 69.2 |
| MInference | 71.7 | 77.6 | 62.3 | 67.9 | 55.2 | 59.8 | 63.1 | 68.4 |
| FlexPrefill | 71.4 | 77.4 | **62.6** | 68.3 | 53.8 | 57.3 | 62.6 | 67.7 |
| XAttention | **71.9** | **78.8** | 62.6 | **68.5** | **55.7** | **60.3** | **63.3** | **69.1** |

*Table 4.* Quantitative results of applying XAttention to the HunyuanVideo model on the VBench benchmark, using a 5-step full-attention warmup. Higher ($\tau$) yields better fidelity (higher PSNR, higher SSIM, lower LPIPS) at the cost of slightly reduced sparsity (higher density). Both ($\tau$) settings demonstrate high similarity to the full attention baseline.

| XAttn $\tau$ | PSNR ($\uparrow$) | SSIM ($\uparrow$) | LPIPS ($\downarrow$) | Density (%, $\downarrow$) |
|---|---|---|---|---|
| 0.90 | 21.5 | 0.767 | 0.215 | 34.4 |
| 0.95 | 23.5 | 0.822 | 0.155 | 45.5 |

trade-off exists: a higher threshold $\tau$ yields better results but slightly lower sparsity. Nevertheless, both configurations achieve over 50% sparsity.

Figure 3 provides a qualitative comparison of videos generated by the baseline (full attention) and XAttention with different configurations using the first prompt in the VBench set. Without the full attention warmup, the generated video, while still high quality, exhibits minor layout differences compared to the baseline. However, with the 5-step full attention warmup, the video generated by XAttention becomes remarkably similar to the one generated by full attention, preserving both high quality and intricate details. These results demonstrate XAttention's effectiveness in video generation models, a promising and increasingly important application area for LCTMs.

### 3.3. Efficiency Results

We further analyze the efficiency of XAttention on tasks with varying context lengths, comparing it against FlashAttention, MInference, and FlexPrefill. We focus on the prefill stage and measure the attention speedup achieved by XAttention. We also break down the computation time into pattern selection and sparse attention components, contrasting it with other trainingless pattern selection methods.

**Attention Acceleration** Figure 4 illustrates the prefill speedup of XAttention across token sequence lengths rang-

*Table 5.* Density on Different Context Lengths. Stride $S = 8$ achieves lower sparsity, and as context length increases, sparsity generally increases (lower density).

| SeqLen | Stride 4 | Stride 8 | Stride 16 |
|---|---|---|---|
| 4k | 51.73% | 52.16% | 55.38% |
| 8k | 40.96% | 43.77% | 43.55% |
| 16k | 27.43% | 27.49% | 28.91% |
| 32k | 21.09% | 20.97% | 27.93% |
| 64k | 9.43% | 10.98% | 11.32% |
| 128k | 6.20% | 6.89% | 7.32% |

ing from 8k to 256k. We conduct these experiments with strides of $S = 16$ and $S = 8$, and a threshold of $\tau = 0.9$. On shorter contexts, where attention density tends to be higher, both MInference and FlexPrefill experience increased overhead due to more extensive pattern selection. In contrast, XAttention maintains its speedup advantage. Notably, for a context length of 256k, XAttention achieves a maximum prefill attention speedup of **13.5x** and **9.8x** with corresponding densities of 7.32% and 6.89%, respectively (see Table 5).

**End to End Speed Up** We further evaluate the end-to-end prefill acceleration of XAttention on the RULER benchmark using Llama-3.1-8B-Instruct. Figure 5 illustrates the trade-off results: The left plot shows how varying sparsity levels affect the proportion of accuracy retained, while the right plot visualizes the relationship between end-to-end speedup and model accuracy. XAttention consistently surpasses baseline methods across both stride = 8 and stride = 16 settings, demonstrating superior efficiency–accuracy trade-offs.

*Table 6.* XAttention End-to-End Speedup on Llama-3.1-8B-Instruct on RULER.

| RULER | 8k | 16k | 32k | 64k | 128k | 256k |
|---|---|---|---|---|---|---|
| XAttn S=8 | 2.59 | 3.04 | 3.96 | 4.38 | 4.67 | 4.93 |
| XAttn S=16 | 2.89 | 3.60 | 4.40 | 4.82 | 4.94 | 5.12 |

**Attention Time Breakdown** Figure 6 demonstrates that XAttention's antidiagonal pattern, coupled with its efficient block selection algorithm, results in significantly faster pattern selection compared to MInference and FlexPrefill, which rely on vertical slash index search. Specifically, XAttention's pattern selection time is up to 24.9x and 5.9x faster, respectively. Furthermore, the accuracy of the antidiagonal pattern allows XAttention to achieve a lower attention density, leading to substantial speedups in the sparse attention computation itself.

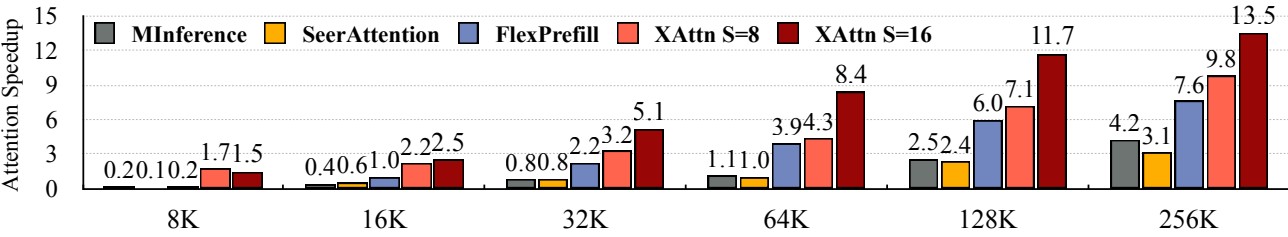

*Figure 4.* Speedup comparison of attention methods across context lengths, relative to FlashInfer's implementation of FlashAttention. XAttention consistently outperforms other sparse attention methods, achieving up to 13.5x speedup at 256K tokens.

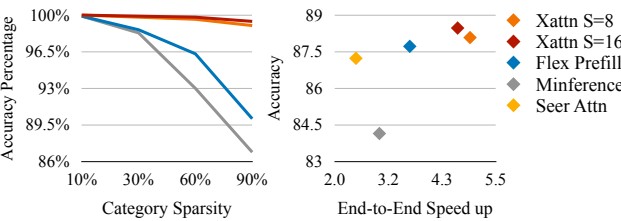

*Figure 5.* Trade-off of XAttention on LLaMA-3.1-8B evaluated on RULER benchmark. The left plot shows the curve between sparsity and the percentage of accuracy retained, while the right plot presents the distribution of end-to-end speedup versus model accuracy. XAttention consistently outperforms baselines at both stride = 8 and stride = 16 settings.

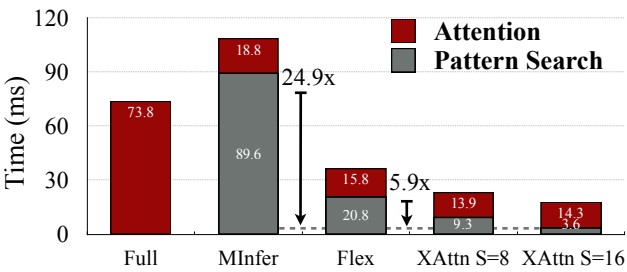

*Figure 6.* Breakdown of prefill attention time. Xattention significantly reduces pattern selection time while maintaining density, achieving substantial acceleration compared to existing methods.

### 3.4. Ablation Study

To further analyze the components of XAttention, we conduct an ablation study, evaluating the effectiveness of the Antidiagonal Pattern, Threshold Block Selection, and Minimum Threshold Prediction.

**Antidiagonal Pattern** We investigate the importance of the antidiagonal pattern by comparing it with random and diagonal patterns as guidance for predicting attention block sums. For the random pattern, we ensure that $S$ elements are selected within each $S \times S$ block, maintaining at least

*Table 7.* Comparison of different patterns. For the same computation, the antidiagonal achieves the lowest density and the highest score.

| Metric | Stride $S = 8$ | | | Stride $S = 16$ | | |
|---|---|---|---|---|---|---|
| | 32k | Avg. | Density | 32k | Avg. | Density |
| Random | 82.53 | 82.48 | 27.57% | 82.35 | 80.94 | 31.36% |
| Diagonal | 76.47 | 81.06 | 24.47% | 58.26 | 79.63 | 25.31% |
| **Antidiagonal** | **90.75** | **88.47** | 20.97% | **90.64** | **88.08** | 27.93% |

one token selection per row and column. Table 7 shows that the antidiagonal pattern achieves the highest accuracy while maintaining the lowest density across tasks, confirming its superiority.

**Stride Sizes** We explore the impact of different stride sizes, $S$. Larger strides lead to sparser sampled attention maps and thus lower computational overhead. However, excessively large strides can compromise the accuracy of block selection. We compare strides of 4, 16, and 64 in Table 8. Our results indicate that when the stride is too long, it fails to accurately detect the previously identified slash attention pattern. An overly sparse antidiagonal cannot effectively distinguish slash patterns entering blocks from different positions, leading to performance degradation.

*Table 8.* Comparison of different Strides. Excessively long strides fail to distinguish slash patterns with different lengths, leading to decreased accuracy.

| Stride | $S = 4$ | $S = 8$ | $S = 16$ | $S = 64$ |
|---|---|---|---|---|
| Avg | 88.89 | 88.47 | 88.08 | 81.21 |
| Density | 21.09% | 20.97% | 27.93% | 39.88% |

**Top-K vs. Top-Ratio vs. Dynamic Sparsity** We evaluate different block selection strategies: Top-K, Top-Ratio, and our Threshold Block Selection (Dynamic Sparsity). For a fair comparison, we set $K = 8192$ and Ratio = 27% for $S = 8$, and $K = 16384$ and Ratio = 31% for $S =$

16, targeting computational costs similar to our Threshold Block Selection. Table 9 demonstrates that both Top-K and Top-Ratio struggle to handle diverse and dynamic input sequence lengths with comparable computation. In contrast, our threshold-based approach, which retains blocks with at least the threshold-level attention, achieves the optimal balance between computation and accuracy.

*Table 9.* Comparison of different selection algorithms.

| Stride | $S = 4$ | | $S = 8$ | | $S = 16$ | |
|---|---|---|---|---|---|---|
| Metric | Avg | Density | Avg | Density | Avg | Density |
| Top K | 84.96 | 17.40% | 84.13 | 19.92% | 83.11 | 30.15% |
| Ratio | 85.96 | 21.00% | 85.42 | 21.00% | 84.24 | 27.00% |
| **Threshold** | **88.89** | 21.09% | **88.47** | 20.97% | **88.08** | 27.93% |

**Minimum Threshold Prediction**   Finally, we compare the performance of our Minimum Threshold Prediction method against a fixed threshold of $\tau = 0.9$ on the RULER benchmark (Hsieh et al., 2024). Using Minimum Threshold Prediction, we start with $\tau = 0.9$ and set $M = 1000$, allowing the dynamic programming (DP) algorithm to explore 1,000 optimal threshold combinations. This results in a set of more refined thresholds, with an average value of 0.8. Table 10 demonstrates that the dynamically predicted threshold achieves lower density and improved accuracy, showcasing the effectiveness of this method.

*Table 10.* Minimum Threshold Prediction yields improvements in both accuracy and sparsity, translating to faster inference.

| Stride | $S = 4$ | | $S = 8$ | | $S = 16$ | |
|---|---|---|---|---|---|---|
| Metric | Avg | Density | Avg | Density | Avg | Density |
| $\tau = 0.9$ | 87.51 | 23.06% | 84.96 | 26.13% | 85.83 | 28.36% |
| Minimum $\tau$ | **88.89** | **21.09%** | **88.47** | **20.97%** | **88.08** | **27.93%** |

## 4. Related Work

### 4.1. Long-Context Large Language Models

Progress in engineering and algorithms has extended the context length capabilities of Large Language Models (LLMs). Two primary approaches are: (1) compiling large datasets of long texts for continuous pretraining or fine-tuning (Peng et al., 2023; Chen et al., 2023), and (2) leveraging external memory or retrieval-augmented techniques to enhance long-range context processing (Burtsev et al., 2021; Xiao et al., 2024a; Wu et al., 2024). These advancements enable LLMs to handle increasingly complex tasks requiring reasoning over extended sequences.

### 4.2. Sparse Attention

The attention mechanism at the heart of LLMs exhibits inherent sparsity, meaning many attention weights are negligible and can be pruned without significant performance degradation (Child et al., 2019a). This sparsity becomes more pronounced as context length increases, presenting opportunities for optimizing inference speed. However, the dynamic and input-dependent nature of this sparsity, which varies across different inputs, attention heads, and even layers, poses a significant challenge for effective exploitation.

Methods like Sparse Transformer (Child et al., 2019b), LongFormer (Beltagy et al., 2020), BigBird (Zaheer et al., 2020) and Selective Attention (Leviathan et al., 2024) reduce complexity through local or block-based attention, but often require retraining, limiting practicality. H2O (Zhang et al., 2023) and TOVA (Oren et al., 2024) discard tokens based on query patterns. StreamingLLM (Xiao et al., 2023b) retains initial and recent tokens for consistent latency and memory usage, enabling processing of sequences longer than the pretraining length. Quest (Tang et al., 2024) uses query-aware token criticality estimation to load only important KV cache pages, accelerating long-context LLM decoding. Retrieval head-based methods (Wu et al., 2024; Xiao et al., 2024b) accelerate model decoding by focusing compute on crucial retrieval heads.

To accelerate the prefill stage, recent methods have employed sparse attention patterns. MInference (Jiang et al., 2024) and FlexPrefill (Lai et al., 2025) both utilize pattern selection algorithms to achieve significant speedups during prefill. However, the overhead of these selection algorithms remains a bottleneck. SeerAttention (Gao et al., 2024) achieves high sparsity through pretraining and fine-tuning of gate parameters, improving efficiency while maintaining low perplexity. Yet, it requires a costly training process and exhibits limited performance on downstream tasks. Therefore, a training-free approach with a minimal-overhead selection algorithm is needed to address the increasingly long prefill times associated with growing context lengths.

### 4.3. LLM Inference Acceleration

Numerous techniques have been developed to accelerate LLM inference. System-level solutions focus on optimizing the original attention computation to better leverage hardware features. Notable examples include FlashAttention (Dao et al., 2022; Dao, 2023), which optimizes memory access patterns for faster attention computation, and RingAttention (Liu et al., 2023), which distributes the attention computation across multiple devices. Other system-level approaches include FlashDecoding (Hong et al., 2024) and PagedAttention (Kwon et al., 2023), which focus on optimizing the computation process and KV cache management, respectively. APE (Yang et al., 2025) aligns parallel

and sequential attention for faster context-augmented generation. Model compression techniques, such as quantization, are also widely employed to reduce model size and memory footprint, leading to faster inference. Examples include SmoothQuant (Xiao et al., 2023a), AWQ (Lin et al., 2024), and QServe (Lin* et al., 2024), which quantize model weights and/or activations to lower bit-widths, thereby reducing memory bandwidth requirements and accelerating computation.

### 4.4. Recent Works

Recently, several outstanding works have focused on advancing sparse attention. Sparse Video Gen (Xi et al., 2025) accelerates video generation models by leveraging spatial and temporal heads while preserving generation quality. NSA (Yuan et al., 2025) introduces a natively trainable sparse attention mechanism for efficient long-context modeling. MoBA (Lu et al., 2025) addresses the quadratic complexity of traditional attention mechanisms without relying on strongly biased structures such as sink or window attention by adopting a Mixture of Experts approach. Fast Video Generation (Zhang et al., 2025) reduces computation demands through Sliding Tile Attention, which employs localized spatial-temporal windows instead of full attention computation. Our work aligns with these efforts to democratize AI by reducing computational costs and enabling efficient deployment.

## 5. Conclusion

We present XAttention, a novel plug-and-play framework for accelerating long-context inference in Transformer models. By leveraging the insight that antidiagonal sums in the attention matrix serve as a robust proxy for block importance, XAttention efficiently identifies and prunes non-essential blocks, achieving substantial computational savings without sacrificing accuracy. Our evaluations on challenging long-context benchmarks in natural language understanding (RULER, LongBench), video understanding (VideoMME), and video generation (VBench) demonstrate that XAttention achieves up to 13.5x speedup in attention computation while maintaining performance comparable to full attention. These results highlight XAttention's ability to unlock the practical potential of block sparse attention, paving the way for efficient and scalable deployment of Long-Context Transformer Models in real-world applications.

## Impact Statement

This paper introduces XAttention, a novel approach for accelerating inference in Long-Context Transformer Models (LCTMs). While the primary goal of this work is to advance

the efficiency of machine learning, particularly in the domain of natural language and video processing, potential societal consequences warrant consideration.

The increased efficiency afforded by XAttention could enable the deployment of LCTMs in resource-constrained environments, broadening access to advanced AI technologies. This could have positive implications for fields like education, healthcare, and accessibility, where LCTMs can be used for tasks such as personalized tutoring, medical diagnosis support, and real-time language translation.

However, the increased accessibility and efficiency of LCTMs also raise potential concerns. These include the potential for misuse in generating misleading or harmful content, exacerbating existing biases in training data, and the potential impact on employment in certain sectors. Furthermore, the ability to process and understand longer contexts could raise privacy concerns if not handled responsibly.

We believe that the benefits of more efficient LCTMs outweigh the potential risks, particularly when developed and deployed responsibly. We encourage researchers and practitioners to consider these ethical implications and to develop safeguards against potential misuse. Further research into the societal impact of efficient LCTMs is crucial for ensuring their beneficial deployment. We hope that XAttention contributes to a future where powerful AI technologies are both accessible and used responsibly for the betterment of society.

ACKNOWLEDGMENTS

We thank MIT-IBM Watson AI Lab, MIT and Amazon Science Hub, MIT AI Hardware Program, National Science Foundation, Hyundai, and Samsung for supporting this research. We thank NVIDIA for donating the DGX server.

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

*Table 11.* Quantitative results of applying XAttention to the Wan2.1 model on the VBench benchmark, using a 5-step full-attention warmup. Higher ($\tau$) yields better fidelity (higher PSNR, higher SSIM, lower LPIPS) at the cost of slightly reduced sparsity (higher density). Both ($\tau$) settings demonstrate high similarity to the full attention baseline.

| XAttn $\tau$ | PSNR ($\uparrow$) | SSIM ($\uparrow$) | LPIPS ($\downarrow$) | Density (%, $\downarrow$) |
|---|---|---|---|---|
| 0.90 | 21.2 | 0.745 | 0.231 | 39.2 |
| 0.95 | 22.7 | 0.819 | 0.129 | 33.6 |

## A. Video Generation Results on Wan 2.1

We also evaluated XAttention on the Wan2.1 model. The resulting accuracy and sparsity metrics are as shown in Table 11.

## B. Various Language Model results on Ruler

*Table 12.* Generalization of XAttention across various model architectures, including Mistral Nemo 12B, Phi-3.5 Mini 3.8B, and Qwen2.5 7B. The results confirm XAttention's sustained effectiveness.

| Model | Method | Average (4k–128k) / Delta |
|---|---|---|
| Mistral Nemo 12B | Full | 67.97 / – |
| | MInference | 64.49 / -3.48 |
| | FlexPrefill | 64.61 / -3.36 |
| | **XAttn S=4** | **67.92 / -0.05** |
| | XAttn S=16 | 67.47 / -0.50 |
| Phi 3.5 Mini 3.8B | Full | 84.68 / – |
| | MInference | 81.89 / -2.79 |
| | FlexPrefill | 82.83 / -1.85 |
| | **XAttn S=4** | **84.86 / +0.18** |
| | XAttn S=16 | 83.82 / -0.86 |
| Qwen2.5 7B | Full | 77.84 / – |
| | MInference | 74.02 / -3.82 |
| | FlexPrefill | 75.10 / -2.74 |
| | **XAttn S=4** | **77.75 / -0.09** |
| | XAttn S=16 | 77.21 / -0.63 |

