# OpenReview forum: "XAttention: Block Sparse Attention with Antidiagonal Scoring"
_ICML.cc/2025/Conference — ICML 2025 poster_

### Official Review · Reviewer_ZhZQ · 2025-02-27

**Overall Recommendation:** 3

**Summary:**

This paper introduces a plug-and-play attention sparsity method with minimal additional computational overhead. The proposed approach uses the sum of antidiagonal values in the attention matrix as a proxy to determine block importance, enabling block selection to reduce the computational density of attention mechanisms. Extensive experiments demonstrate that the method significantly enhances attention sparsity and accelerates attention computation, all while maintaining the model's performance.

## update after rebuttal
While the rebuttal has addressed my main concerns, I agree with reviewer 5Q7x that the technical mechanism appears incremental. So I keep my score (3).

**Claims And Evidence:**

Yes

**Essential References Not Discussed:**

No

**Experimental Designs Or Analyses:**

There are several issues:
1. There is a lack of sufficient justifications for why the selected baselines are the most appropriate for evaluating the proposed method.
2. The experiments only test a narrow range of stride values, and the paper does not provide a clear strategy for determining stride. This may limit the generalizability of the method, as stride selection could significantly impact performance.

**Methods And Evaluation Criteria:**

There are several issues:
1. The proposed method relies on the antidiagonal value-based block importance estimation, and the paper claims that the antidiagonal selection method is effective due to its advantages in Information Preservation and Pattern Detection. However, these advantages are not unique to antidiagonal values, as the main diagonal can also preserve information and detect patterns. Further theoretical or empirical evidence is required to clarify the rationale and necessity behind the antidiagonal selection strategy.
2. The method relies on a precomputed attention matrix to predict block importance, but the paper does not clearly explain when and how this step is performed.
3. How the Minimum Threshold Prediction works is not that clear. The paper should explicitly explain the definition of a time of threshold adjustment and when the threshold is adjusted, especially during the prefilling process.

**Other Comments Or Suggestions:**

None

**Other Strengths And Weaknesses:**

The paper addresses an important problem and shows promise, but the motivation and justification for the proposed method are currently unclear.

**Questions For Authors:**

1. What are the unique motivations for using antidiagonal values to estimate block importance?
2. If the sparsity level is predefined, how can the proposed method achieve a predefined sparsity?

**Relation To Broader Scientific Literature:**

Different from existing works that rely on pooling methods, the paper proposed an antidiagonal selection strategy to achieve attention sparsity, which introduces less additional computational overhead.

**Theoretical Claims:**

There is no theoretical claim.

---

> ### Author Rebuttal · Authors · 2025-03-29
>
> ### 1. Antidiagonal Selection: Performance and Insights
>
> Antidiagonal selection offers significant advantages over other patterns:
>
> - Retains **all token information** while simultaneously detecting both **vertical and slash patterns** critical in LLM prefill
> - Diagonal patterns **miss** slash patterns with probability $\frac{\text{stride} - 1}{\text{stride}}$
> - Other patterns (horizontal lines, pooling methods) lose token-level information
> - Empirically validated superiority: outperforms diagonal patterns (Table 6) and pooling-based methods used by FlexPrefill
>
> ### 2. Precomputation Process
>
> Precomputation occurs before computing the full attention for $Q\times K^T$. Based on block selection results, we determine which blocks require full attention computation. The process is detailed in Algorithm 1.
>
> ### 3. Minimum Threshold Prediction
>
> Minimum Threshold Prediction is an **optional algorithm** which finds the optimal threshold for each head, further optimizing efficiency and accuracy. This process:
>
> - Is conducted **offline** (not during prefill)
> - Does not increase runtime computational complexity
> - Profiles configurations in advance, determining minimum thresholds per head
>
> ### 4. Baseline Selection Justification
>
> Our selected baselines represent SOTA methods for sparse prefill attention. Previous approaches:
>
> - Have high precomputation complexity and overhead (Figure 5)
> - Limited applicability (FlexPrefill/MInference don't support video generation or chunk prefill)
> - Require additional fine-tuning (SeerAttention)
>
>
> ### 5. Stride Selection Strategy
>
> Stride is indeed critical for balancing efficiency and accuracy:
>
> **Different stride values (tested on RULER with Llama3.1 8B):**
>
> | Stride   | 64    | 32    | 16    | 8     | 4     | 2     | 1     |
> | -------- | ----- | ----- | ----- | ----- | ----- | ----- | ----- |
> | Overhead | 1.97% | 3.12% | 4.81% | 7.54% | 14.9% | 28.9% | 57.4% |
> | Accuracy | 81.21 | 84.23 | 88.04 | 88.42 | 88.38 | 88.39 | 88.27 |
>
> These results show stride = 8 maintains accuracy comparable to stride = 4 with less overhead, and performance remains effective for stride ≤ 16.
>
> According to our complexity analysis:
>
> - Precomputation: $\frac{2n^2d}{\text{stride}} + \frac{3n^2}{\text{stride}^2}$
> - Full attention: $4n^2d + 4n^2 + 6nd^2$
>
> With stride = 4, precomputation cost is < 1/8 of full compute
>  With stride = 8, precomputation cost is < 1/16 of full compute
>
> (As shown in Figure 5, while the actual latency closely aligns with theoretical predictions, it may be slightly higher due to factors such as vector reordering, memory overhead, and the implementation details of the Triton kernel.)
>
> ### 6. Predefined Sparsity Implementation
>
> Fixed sparsity can be achieved by setting a block limit k and selecting the top-k highest-scoring blocks. This approach is equivalent to the Top-k strategy (Table 8).
>
> However, we believe fixed sparsity across all inputs is suboptimal since information density varies between requests. XAttention's **dynamic sparsity determination** enables better generalization and accuracy across diverse scenarios.

---

> > ### Comment · Reviewer_ZhZQ · 2025-04-02
> >
> > Thanks for the rebuttal. It addresses my main concerns.

---

> > > ### Author Response · Authors · 2025-04-02
> > >
> > > Thank you for confirming that your main concerns have been addressed. Please don’t hesitate to let us know if you have any further questions—we’d be more than happy to clarify or provide additional details. If there are no remaining concerns, we would greatly appreciate it if you could consider updating your evaluation to reflect your current view of the paper.
> > >
> > > Very best,
> > >
> > > Authors

---

### Official Review · Reviewer_F2rz · 2025-03-10

**Overall Recommendation:** 4

**Summary:**

This paper introduces XAttention, a plug-and-play framework that accelerates long-context inference in Transformer models through block sparse attention. The key innovation is using the sum of antidiagonal values in the attention matrix as a proxy for block importance, allowing for identification and pruning of non-essential blocks. This approach achieves considerable speedups (up to 13.5x) while maintaining comparable accuracy to full attention across language, video understanding, and video generation tasks. The method doesn't require retraining and shows promising results compared to other block-sparse approaches like MInference, FlexPrefill, and SeerAttention.

**Claims And Evidence:**

+The claims made in this submission are generally supported by convincing evidence. The main claim of achieving significant speedups while maintaining accuracy is well-supported by experimental results across multiple benchmarks.

-However, the evidence is somewhat limited in model diversity, with primary evaluations on Llama-3.1-8B-Instruct for language tasks, Qwen2-VL-7B-Instruct for video understanding, and HunyuanVideo for video generation. The efficacy of the approach across model architectures beyond these specific models is not fully established, which weakens the generality claim somewhat. It would test more models, e.g., Mistral Nemo 12B Instruct, Phi 3.5 Mini 3.8B Instruct, Qwen2.5 7B Instruct, and so on.

**Essential References Not Discussed:**

NA

**Experimental Designs Or Analyses:**

+The experimental design is generally sound. The authors evaluate on diverse tasks (language, video understanding, video generation) with varying sequence lengths (4k to 256k tokens), providing a comprehensive picture of the method's capabilities.

-One issue is the relatively simple setup for video generation experiments. While the comparison against full attention is done using the same random seed and prompts, only PSNR, SSIM, and LPIPS metrics are reported without detailed analysis of generation quality or more nuanced evaluation. More qualitative analysis would be beneficial.

**Methods And Evaluation Criteria:**

+The paper uses established benchmarks (RULER, LongBench, VideoMME, and VBench) that directly test the challenges of long-context understanding, which align well with the goal of the work. The comparison against strong baselines (FlashAttention, MInference, FlexPrefill, SeerAttention) provides good context for understanding the gains.

+The ablation studies are valuable in examining the contributions of different components (antidiagonal pattern, threshold block selection, minimum threshold prediction), though more exploration of hyperparameter sensitivity would strengthen the evaluation.

**Other Comments Or Suggestions:**

NA

**Other Strengths And Weaknesses:**

Strengths:
- The antidiagonal scoring idea is novel and interesting - it provides a simple yet effective heuristic for identifying important attention blocks without requiring expensive computation.
- The speedup achieved (up to 13.5x for 256k context) is impressive. Figure 4 clearly shows how XAttention consistently outperforms other sparse attention methods across different sequence lengths.
- The method is training-free and can be applied as a drop-in replacement, making it immediately useful for practitioners without requiring costly retraining or fine-tuning. The dynamic threshold prediction approach shows thoughtful consideration of the varying sparsity patterns across different attention heads.

Weaknesses:
- Limited model diversity - the evaluation focuses primarily on Llama-3.1-8B-Instruct for language tasks, with limited exploration across model families or sizes. This raises questions about how well the approach generalizes across different model architectures and scales.
- The video generation experiments, while novel, feel preliminary. More detailed analysis beyond basic metrics would strengthen these results.

**Questions For Authors:**

NA

**Relation To Broader Scientific Literature:**

The authors connect their approach to previous work on sparse attention (Sparse Transformer, LongFormer, BigBird, etc.) and more recent work on attention optimizations like FlashAttention and inference acceleration methods. The paper clearly identifies its novelty compared to related approaches like MInference and FlexPrefill, highlighting that those methods incur significant computational overhead for pattern selection, which XAttention addresses through its antidiagonal scoring technique.

**Theoretical Claims:**

The paper does not have theory.

---

> ### Author Rebuttal · Authors · 2025-03-29
>
> ### 1. Model Generalizability
>
> We tested XAttention across diverse architectures with consistent results:
>
> **LLMs Accuracy (RULER)**
>
> | Model             | Method      | Average (4k-128k) | Performance Delta |
> | ----------------- | ----------- | ----------------- | ----------------- |
> | Mistral Nemo 12B  | Full        | 67.97             | –                 |
> |                   | MInference  | 64.49             | -3.48             |
> |                   | FlexPrefill | 64.61             | -3.36             |
> |                   | XAttn S=4   | **67.92**         | **-0.05**         |
> |                   | XAttn S=16  | 67.47             | -0.50             |
> | Phi 3.5 Mini 3.8B | Full        | 84.68             | –                 |
> |                   | MInference  | 81.89             | -2.79             |
> |                   | FlexPrefill | 82.83             | -1.85             |
> |                   | XAttn S=4   | **84.86**         | **+0.18**         |
> |                   | XAttn S=16  | 83.82             | -0.86             |
> | Qwen2.5 7B        | Full        | 77.84             | –                 |
> |                   | MInference  | 74.02             | -3.82             |
> |                   | FlexPrefill | 75.10             | -2.74             |
> |                   | XAttn S=4   | **77.75**         | **-0.09**         |
> |                   | XAttn S=16  | 77.21             | -0.63             |
>
> **LLMs Speed-up**
>
> | Model        | Method     | 8k       | 16k      | 32k      | 64k      | 128k      | 256k      |
> | ------------ | ---------- | -------- | -------- | -------- | -------- | --------- | --------- |
> | Mistral Nemo | XAttn S=16 | **1.7×** | **2.6×** | **4.7×** | **8.3×** | **9.9×**  | **10.9×** |
> | Phi 3.5      | XAttn S=16 | 1.4×     | **2.3×** | **4.7×** | **8.1×** | **11.0×** | **11.9×** |
> | Qwen2.5 7B   | XAttn S=16 | **1.8×** | **2.7×** | **4.9×** | **8.5×** | **12.4×** | **13.9×** |
>
> **Video Generation (Wan 2.1 14B)**
>
> | Threshold | PSNR (↑) | SSIM (↑) | LPIPS (↓) | Density (%) | Speed-up |
> | --------- | -------- | -------- | --------- | ----------- | -------- |
> | 0.90      | 21.20    | 0.739    | 0.212     | 35.8        | 2.5×     |
> | 0.95      | 22.67    | 0.794    | 0.167     | 51.2        | 1.8×     |
>
> ---
>
> ### 2. Hyperparameter Sensitivity
>
> Threshold = 0.9 demonstrates strong **generalizability** across models:
>
> **Same threshold (0.9) on different models:**
>
> | Model         | Llama  | Mistral Nemo | Phi3.5-mini | Qwen2.5 |
> | ------------- | ------ | ------------ | ----------- | ------- |
> | Sparsity      | 23.06% | 24.93%       | 29.32%      | 21.15%  |
> | Performance Δ | -0.01  | -0.05        | +0.18       | -0.09   |
>
> **Trade-off curve with LLama 3.1 8B (stride=4):**
>
> | Threshold | 0.1   | 0.7   | 0.8    | 0.9    | 0.95   | 1.0     |
> | --------- | ----- | ----- | ------ | ------ | ------ | ------- |
> | Sparsity  | 4.31% | 5.62% | 10.35% | 23.06% | 49.88% | 100.00% |
> | Accuracy  | 41.34 | 73.96 | 84.39  | 87.51  | 87.64  | 87.52   |
>
> ---
>
> ### 3. Video Generation Quality
>
> These quantitative metrics are well-acknowledged methods used in efficient visual content generation works to compare the difference between original generated content and the efficient generated content, such as used in Distrifusion (Li et al., https://arxiv.org/pdf/2402.19481) and Sparse VideoGen (Xi et al., https://arxiv.org/pdf/2502.01776). They are extremely strict compared to content level scores since they require pixel-level exactness, and the scores XAttention achieved is a level which very similar to the original generation. Beyond quantitative metrics, we've included qualitative samples in the Figure 3 comparing between XAttention-generated videos and full attention videos, confirming they are virtually indistinguishable.

---

> > ### Comment · Reviewer_F2rz · 2025-04-02
> >
> > Thank you for your response. The new results seem strong and fixed my concerns. Thus, I will raise my score from 3 to 4.

---

> > > ### Author Response · Authors · 2025-04-02
> > >
> > > Thank you for your thoughtful confirmation and for raising your score. We're glad the new results addressed your concerns, and we truly appreciate your updated evaluation!
> > >
> > > Very best,
> > >
> > > Authors

---

### Official Review · Reviewer_aVq1 · 2025-03-12

**Overall Recommendation:** 3

**Summary:**

- This paper proposes an efficient attention model.
- It finds out that the antidiagonal values in the attention matrix provides a powerful proxy for block importance
- Unlike existing methods that primarily rely on computationally intensive and lossy solutions
like token pooling to identify important blocks, Xattention directly use attention scores which are more efficient.
- The proposed module is tested on several benchmarks including language process, video understanding and video generation.

**Claims And Evidence:**

- Not exactly,
- Although extensive experiments are provided. The algorithm itself is confusing, the implementation details, calculation equations, theoretical computational complexity are not provided to help understand this method.
- The motivation and design choice of threshold prediction and dynamic programming are confusing.
- The method achieves both notable performance and efficiency improvements compared to previous methods, but the underlying reason remains unclear.

**Essential References Not Discussed:**

None.

**Experimental Designs Or Analyses:**

Yes

**Methods And Evaluation Criteria:**

Yes, three different tasks (language process, video understanding and video generation) on several benchmarks are adopted for evaluation.
Efficiency comparison of the proposed attention module is also provided.

**Other Comments Or Suggestions:**

I think the paper writing needs extensive improvement. Especially the algorithm, implementation details, design choice analysis and computational complexity comparison.

**Other Strengths And Weaknesses:**

- Some implementation details of experiments are not provided. For example, does XAttention needs finetuning or it can directly replace standard attention during inference? In Language tasks, does XAttention both works in prefill and decoding stages?
- The figures shown in the paper are confusing, which seem not align with the main algorithm and demand extensive effort to understand.

**Questions For Authors:**

Does this method use any cuda or triton implementation? Ot it uses any existing efficient attention components or code?

**Relation To Broader Scientific Literature:**

None

**Theoretical Claims:**

Yes, the  Algorithm 1 is checked.
- The Algorithm 1 involves two levels of loops to select blocks, what's its computational efficiency and why it is more efficient?
- Why the approximate softmax attention is calculated inside the loops.

---

> ### Author Rebuttal · Authors · 2025-03-29
>
> ### 1. Algorithm Clarification
>
> XAttention precomputes attention within the **antidiagonal pattern** and uses these scores to guide block-sparse attention selection. Our method:
>
> - Requires no fine-tuning
> - Achieves **lowest overhead** among prefill acceleration approaches (Figure 5)
> - Delivers up to **13.5× speedup** over FlashAttention (Figure 4)
>
> ---
>
> ### 2. Complexity Analysis
>
> **Precomputation complexity:**
> $$\frac{2n^2d}{\text{stride}} \quad (\text{approximate attention}) \ + \ \frac{3n^2}{\text{stride}^2} \quad (\text{approximate softmax})$$
>
> **Full attention complexity:**
>  $$4n^2d + 3n^2 + 4nd^2$$
>
> With **stride = 4**, precomputation cost is < **1/8** of full compute
> With **stride = 16**, precomputation cost is < **1/32** of full compute.
>
> **Figure 5** shows a breakdown of precompute time that closely aligns with this theoretical result. (While the actual latency closely aligns with theoretical predictions, it may be slightly higher due to factors such as vector reordering, memory overhead, and the implementation details of the Triton kernel.)
>
> ---
>
> ### 3. Threshold DP Design Choice
>
> As described in Section 2.3, this is an **optional component** for optimizing sparsity by assigning different thresholds per head. XAttention also works effectively with global thresholds on LLMs, VLMs, and video generation.
>
> ---
>
> ### 4. Antidiagonal Pattern Effectiveness
>
> The antidiagonal pattern:
>
> - Retains all token information
> - Simultaneously detects vertical and slash patterns with $1/\text{stride}$ probability
> - Prevents information loss compared to horizontal lines or pooling methods
> - Empirically outperforms both diagonal patterns (Table 6) and pooling methods (e.g., FlexPrefill)
>
> ---
>
> ### 5. Two-Level Loop Structure
>
> In Algorithm 1:
>
> - Outer loop: blocks (`blocknum`)
> - Inner loop: stride slices (`stride`)
>
> Despite appearing as nested loops, the **actual complexity is O(n)**, not O(n²), where n represents sequence length.
>
> ---
>
> ### 6. Softmax Approximation
>
> Line 120 explains why we normalize attention scores - to create a probability distribution for threshold selection. The Softmax complexity here is $\left(\frac{\text{stride}}{\text{blocksize}}\right)^2$ of full attention (e.g., $\frac{1}{1024}$ with stride = 4).
>
> Top-k could replace Softmax, but Table 8 demonstrates this performs worse than our threshold method.
>
> ---
>
> ### 7. Implementation Details
>
> As noted in line 20 of the abstract, our method is:
>
> - **Plug-and-play**
> - Requires **no finetuning**
> - **Specifically designed for prefill stage**
>
> We used both Block Sparse Attention (https://github.com/mit-han-lab/Block-Sparse-Attention) CUDA kernel and Triton kernel for implementation.
>
>
> ---
>
> ### 8. Figure Presentation
>
> Thank you for these suggestions. We will improve figure clarity in the revision.

---

> > ### Comment · Reviewer_aVq1 · 2025-04-07
> >
> > Thank you for the response, after carefully reading the rebuttal, code and paper, some of the algorithm do make sense.
> > I will raise my score accordingly. But I am still wondering:
> > - Do the important blocks calculated with the the antidiagonal pattern statistically align with the importance blocks calculated with original attention scores or pooled attention scores?

---

> > > ### Author Response · Authors · 2025-04-09
> > >
> > > Thank you for taking the time to carefully review our rebuttal, and paper. We greatly appreciate your willingness to raise your score based on the deeper understanding of our algorithm.
> > >
> > > You raised a great question. We conducted a thorough statistical analysis to address this point.
> > >
> > > First, we evaluated the Spearman's rank correlation between various scoring strategies and the original full attention map scores across 100 random inputs with sequence length of 4K:
> > >
> > > | Method        | Antidiagonal | Diagonal | Sum Pooling | Max Pooling |
> > > | ------------- | ------------ | -------- | ----------- | ----------- |
> > > | Correlation ↑ | 0.49         | 0.32     | 0.14        | 0.01        |
> > >
> > > While revealing, correlation alone doesn't fully capture our algorithm's objective. Since XAttention only needs to identify which blocks to compute (not their exact ranking), we further analyzed using Area Under the ROC Curve (AUC) - a more suitable metric for binary classification tasks.
> > >
> > > For this analysis, we:
> > >
> > > 1. Labeled blocks from the full attention map exceeding the threshold as positive (1), others as negative (0)
> > > 2. Used scores from each method as predictions
> > > 3. Calculated AUC to measure each method's ability to correctly classify important blocks
> > >
> > > | Method      | Antidiagonal | Diagonal | Sum Pooling | Max Pooling |
> > > | ----------- | ------------ | -------- | ----------- | ----------- |
> > > | AUC Score ↑ | 0.84         | 0.69     | 0.55        | 0.52        |
> > >
> > > The antidiagonal pattern significantly outperforms other methods in both metrics. An AUC of 0.84 indicates strong discriminative ability - the antidiagonal scores successfully identify ~84% of the truly important blocks that would be selected using full attention scores.
> > >
> > > This superior performance aligns with our theoretical analysis: the antidiagonal pattern efficiently captures both vertical and slash patterns while preserving token-level information, making it an excellent proxy for identifying attention hotspots.
> > >
> > > We're open to any additional questions you might have about our work.

---

### Official Review · Reviewer_5Q7x · 2025-03-14

**Overall Recommendation:** 3

**Summary:**

This paper introduces XAttention, a novel block-sparse attention mechanism leveraging an "antidiagonal scoring" method to efficiently approximate standard transformer attention. XAttention aims to accelerate inference in Long-Context Transformer Models (LCTMs) by using an antidiagonal scoring strategy to identify and prune less important blocks within the attention matrix. The authors argue that summing antidiagonal values within blocks effectively captures critical attention regions, allowing for computational savings without substantial accuracy loss. Empirical evaluations on multiple benchmarks (RULER, LongBench, VideoMME, VBench) demonstrate that XAttention achieves competitive accuracy compared to full attention methods while substantially reducing computational overhead, showing speedups up to 13.5×.

**Claims And Evidence:**

Yes

**Essential References Not Discussed:**

The following essential reference about block-wise attention is not discussed in this paper:

[1] Blockwise Self-Attention for Long Document Understanding, ACL 2020

**Experimental Designs Or Analyses:**

Yes

**Methods And Evaluation Criteria:**

Yes

**Other Comments Or Suggestions:**

The spacing between the captions for tables and figures and the surrounding text appears insufficient, which could affect both readability and the overall presentation. It is recommended that additional spacing be introduced to enhance the visual clarity of the document. For example:

1. The caption for Figure 2 (lines 66–68)
2. The caption for the table (lines 165–169)
3. Table 3 (lines 279–280)
4. The caption in Table 4 (lines 294–295)

**Other Strengths And Weaknesses:**

Strengths:

1. The observation that antidiagonal values within the attention matrix can serve as a powerful indicator of block importance makes sense.

2. Clearly motivated method with an efficient antidiagonal scoring strategy.

3. Extensive and solid experiments across diverse benchmarks (text and video domains), providing good empirical support.

4. Demonstrates substantial computational efficiency with competitive accuracy relative to dense attention methods.

Weaknesses:

1. While the paper presents some refinements in block sparse attention methods compared to existing approaches such as [1], the improvements appear to be incremental. It would be valuable for the authors to elaborate on how antidiagonal selections offer significant performance.

2. While the use of the antidiagonal as an importance indicator for the block is an interesting idea, it appears to be the primary novel contribution of the work. It appears that the improvements made to the methods are somewhat limited.

3. Lack of deeper analyses and insights to justify why antidiagonal scoring outperforms other potential scoring methods compared to the other pooling methods. It would be beneficial if the authors could provide additional theoretical insights to further substantiate and highlight the advantages of this approach compared to existing methods.

4. Lack of a speed comparison between Xattention and FlashAttention.


[1] Blockwise Self-Attention for Long Document Understanding, ACL 2020

**Questions For Authors:**

Could you please explain why the appendices have not been included?

**Relation To Broader Scientific Literature:**

The key contributions of this paper show that the sum of antidiagonal selected elements serves as a proxy for the overall importance of the corresponding attention block, which is related to block-wise sparse attention.

**Theoretical Claims:**

There are no theoretical claims in this work.

---

> ### Author Rebuttal · Authors · 2025-03-29
>
> ### 1. Antidiagonal Selection: Performance and Insights
>
> Antidiagonal selection offers significant advantages over other patterns:
>
> - Retains **all token information** while simultaneously detecting both **vertical and slash patterns** critical in LLM prefill
> - Diagonal patterns **miss** slash patterns with probability $\frac{\text{stride} - 1}{\text{stride}}$
> - Other patterns (horizontal lines, pooling methods) lose token-level information
> - Empirically validated superiority: outperforms diagonal patterns (Table 6) and pooling-based methods used by FlexPrefill
>
> ---
>
> ### 2. Novel Contribution
>
> We respectfully disagree with the assessment of limited contributions. While block-sparse attention exists, **effectively implementing it without accuracy/efficiency degradation remains challenging**:
>
> - FlexPrefill and MInference: Suffer from **high precomputation costs** and rely on last block for pattern detection, preventing **chunk prefill** and adaptation to **non-text tasks** like video
> - SeerAttention: Requires **additional parameter training**, limiting generalizability. Also it has poor empirical performance.
> - Our work: First to identify antidiagonal pattern as an effective block importance proxy, creating a **simple yet powerful method** that significantly improves the efficiency-accuracy trade-off
>
> ---
>
> ### 3. FlashAttention Comparison
>
> As shown in the Figure 4 of our submitted paper, XAttention achieves up to **13.5× speedup** over FlashInfer FlashAttention (one of the fastest implementations):
>
> | Prefill Time (ms)         | 8k   | 16k  | 32k  | 64k   | 128k   | 256k   |
> | ------------------------- | ---- | ---- | ---- | ----- | ------ | ------ |
> | FlashInfer                | 5.0  | 19.3 | 77.2 | 314.5 | 1269.5 | 5192.1 |
> | FlashAttention (Official) | 4.9  | 19.1 | 76.8 | 312.2 | 1265.8 | 5186.5 |
> | XAttn Stride = 4                | 3.3  | 8.2  | 24.4 | 63.2  | 181.4  | 543.6  |
> | XAttn Stride = 16               | 2.6  | 5.2  | 13.4 | 38.3  | 134.0  | 383.6  |
>
>
> ### 4. References and Formatting
>
> Thank you for noting these issues. We will address them in the revision.
>
> ---
>
> ### 5. Appendices
>
> The main paper effectively covers our contributions. We will include an appendix showing additional visual samples, theoretical analysis, and experimental results in the next revision.

---

### Decision · Program_Chairs · 2025-05-01

**Decision:**

Accept (poster)

**Comment:**

This paper presents XAttention, a plug-and-play method for accelerating long-context inference in Transformers via a novel antidiagonal block scoring strategy. The core idea is to use the sum of antidiagonal values in attention blocks as a proxy for block importance, enabling efficient block-sparse attention without sacrificing performance. The authors demonstrate strong empirical gains with competitive accuracy across a diverse set of benchmarks, including language modeling, video understanding, and video generation.

The reviewers generally applauded for the novelty of the antidiagonal scoring mechanism and its simplicity, and are satisfied with the strong empirical support with evaluation across multiple domains (RULER, LongBench, VideoMME, VBench) and architectures (LLaMA-3, Qwen2.5, Phi-3.5, Mistral Nemo), where the method consistently matches or exceeds baseline accuracy while delivering substantial speedups. While there were some initial minor concerns, most of those are cleared through the process of discussion and reviewers unanimously suggested acceptance. Hence I recommend acceptance.